# Fast GC E-Nose and Chemometrics for the Rapid Assessment of Basil Aroma

Lorenzo Strani [1], Alessandro D'Alessandro [1,2], Daniele Ballestrieri [2], Caterina Durante [1,*] and Marina Cocchi [1]

1 Department of Chemical and Geological Sciences, University of Modena and Reggio Emilia, Via Campi 103, 41125 Modena, Italy; lostrani@unimore.it (L.S.); alessandro.dalessandro@barilla.com (A.D.); marina.cocchi@unimore.it (M.C.)
2 Barilla G. e R. Fratelli, Via Mantova 166, 43122 Parma, Italy; daniele.ballestrieri@barilla.com
* Correspondence: cdurante@unimore.it; Tel.: +39-059-2058-554

**Abstract:** The aim of this work is to assess the potentialities of the synergistic combination of an ultra-fast chromatography-based electronic nose as a fingerprinting technique and multivariate data analysis in the context of food quality control and to investigate the influence of some factors, i.e., basil variety, cut, and year of crop, in the final aroma of the samples. A low = level data fusion approach coupled with Principal Component Analysis (PCA) and ANOVA—Simultaneous Component Analysis (ASCA) was used in order to analyze the chromatographic signals acquired with two different columns (MXT-5 and MXT-1701). While the PCA analysis results highlighted the peculiarity of some basil varieties, differing either by a higher concentration of some of the detected chemical compounds or by the presence of different compounds, the ASCA analysis pointed out that variety and year are the most relevant effects, and also confirmed the results of previous investigations.

**Keywords:** basil; aroma; fast GC; electronic nose; untargeted fingerprint; PCA; ASCA; cut; variety

## 1. Introduction

Aromatic herbs of the *Laminacae* family are largely employed worldwide in culinary and health-related uses [1]. Among them, basil is largely used and very appreciated for its distinctive flavour, and its essential oils possess numerous health properties. Basil's origin dates back to over thousands of years; its name seems to derive from the ancient Greek "basilikon" (plant of the king), seemingly given for its peculiar characteristics [2]. There is a large number of basil cultivars and, for this reason, a standardized descriptor list, based on morphological characteristics, was developed by the International Union of Protection of New Varieties Plants (UPOV) [3]. In this list, *O. basilicum* is divided into six distinct morphotypes: 1. purple A, 2. purple B, 3. purple C, 4. lettuce, 5. small leaves, and 6. true basil [4]. Basil flavor is composed of different classes of molecules, such as ketones, alcohols, terpenoids, and esters [5], and for these reasons, a further classification scheme was proposed considering the different chemotypes: 1. high-linalool, 2. linalool/trans-α-bergamotene, 3. linalool/estragole, 4. linalool/trans-methyl cinnamate, and 5. high-estragole [6,7].

Basil has a relevant place in the Italian culinary culture, in the context of which it is largely used and appreciated [8]. There are different basil varieties [9] and are used, for example, in "pesto", a typical green sauce of the Italian region Liguria, in which a linalool/estragole basil chemotype prevails. In the last few years, in Italy, basil demand has increased: from 2015 to 2020, the harvested surface was more than doubled as was the produced quantity [10]. In this context, the selection of new varieties with improved agronomic characteristics and richer in appreciated flavor notes also became a relevant aspect. Traditionally, basil for food industry use has been cultivated on open fields, but greenhouses are sometimes used in early or late crops for productivity reasons. Normally,

in warmer climates, such as Italy, three-to-five cuts per harvesting year can be carried out; the first cut usually begins in late spring or early summer and the following cuts after about 20 days, depending on the weather conditions, and just before or at the start of flowering [11,12].

The development of 'artificial senses' for the evaluation of food quality and consumer preferences is nowadays well established [13]. In fact, on the one hand, they mimic food perceptions, and, on the other hand, they may furnish a quick evaluation and characterization of specific food attributes. In particular, under the general term electronic nose (e-nose) are comprised all types of sensors capable of detecting volatile organic compounds (VOCs), and include optical, electrical, electrochemical and mass-based detection [14,15]. Despite their different mechanisms, most of these sensors show non-specific recognition since they interact non-selectively with volatile molecules. In recent years, a new generation of e-nose instruments, based on ultra-fast gas chromatography with flame ionization detection (FID), i.e., fast GC-enose, has emerged as an appealing technology for VOC detection [16,17]. In fact, it shares the fast-screening capability of other types of e-noses, while allowing, at the same time, specificity and the putative identification of the detected molecules, which can be afterward confirmed by using a chromatographic run with standards or by GC-MS.

In order to characterize the basil flavor pattern, many analytical methods have been developed [18], such as the solvent extraction of the essential oil, or the direct sampling of the released volatile molecules by means of different analytical tools [19]. In fact, the more common tools are based on the direct collection of the headspace, or the trapping of the volatile molecules by Solid-Phase Micro Extraction (SPME) or by Head-Space Sorptive Extraction (HSSE) [20], while for the determination of the essential oil, gas chromatography (GC) is mainly employed, either coupled with mass spectrometry (MS) to have an identification, or just using flame ionization detection (FID), if identification is not the main concern.

In a previous study [21], our team developed an analytical method, based on e-nose ultrafast GC-FID, to characterize the basil flavor profile of some of the varieties currently employed in the production of Italian pesto sauce. Among the more than thirty peaks detected, only eighteen were tentatively identified on the basis of Kovats relative retention indices, and finally nine were confirmed by the analysis of the pure molecules. For these nine molecules, quantification was performed, constructing, for each one, a calibration curve with internal standards. These chemical markers allowed a partial chemical characterization of basil aroma profiles, and a differentiation of basil samples according to the studied agronomic factors.

The possibility to observe the complete chromatogram in an unsupervised way was the natural progression to fully benefit from the potential of the fast GC method. To this aim, in the present paper, the raw chromatographic signals, acquired in a very short time (110 s) by two different GC columns, are integrated according to a low-level data fusion approach [22,23], instead of considering (and quantifying) only the nine a priori known markers and the outcome of a single column. In addition, a higher number of basil samples collected from 2019 to 2021 (this year has not been previously considered) are measured, at the same time that the number of varieties studied is increased. Finally, the focus is the extraction of reliable chemical information from the raw signals aided by proper data analysis and preprocessing tools. In this way, without the need and the effort of the identification and quantification of specific markers, it is possible to study the different factors linked to production aspects and their influence on the product quality. This kind of approach could be easily and rapidly exported to other products where the knowledge of the individual molecules is more challenging or time consuming.

Multivariate data analysis pipeline included proper preprocessing, exploratory analysis by Principal Component Analysis (PCA), and ANOVA Simultaneous Component Analysis (ASCA) [24] to assess the effect of varieties, cuts period and harvesting years (2019, 2020 and 2021) on basil aroma. These are very critical aspects to consider when planning

the basil agronomic campaign in order to control the quality of pesto sauce, which is the product of interest.

## 2. Materials and Methods

### 2.1. Basil Plants

Plants of basil (*Ocimum basilicum*) of 20 commercial varieties of the "Genovese" type have been supplied by local producers over three different harvest years from 2019 to 2021. The varieties name was declared as code for confidentiality reasons and only the "Italiano Classico" variety was clearly indicated due to its largely commercial use. A total of 253 samples were collected and analysed.

Each basil variety was collected at different plant ages indicated as "cut". The plants were cut leaving about 5–6 cm from the soil, to allow the plants to regrow before the next cut. Typically, the first cut (labeled as 1 in Table 1) is performed after about 40 days from sowing and, then, the following cuts (numbered in time order from 2 to 5 in Table 1) after about 20 days each, depending on the weather and the agronomic conditions. Details of all samples (352 in total) are reported in Table 1. Varieties and cuts were not regularly varied during the three years because of company and producer constraints.

**Table 1.** Samples analyzed during the three years with the indication of the number of samples considered for each cut and, in italics, the number of replicates for each sample.

| Harvesting Year | Basil Variety | Cut in Bold (n° of Samples; *Total Replicates*) | | | | |
|---|---|---|---|---|---|---|
| 2019 | Italiano Classico | **1** (5; *18*) | **2** (2; *6*) | **3** (2; *6*) | **4** (2; *6*) | |
| | variety 5 | **1** (1; *3*) | | | | |
| | variety 7 | **1** (2; *9*) | | | | |
| | variety 9 | **1** (1; *5*) | **2** (1; *3*) | **3** (1; *3*) | **4** (1; *3*) | |
| | variety 13 | **1** (2; *3*) | | | | |
| | variety 14 | | **2** (1; *3*) | **3** (1; *2*) | **4** (1; *3*) | |
| | variety 17 | **1** (2; *5*) | **2** (1; *3*) | **3** (1; *3*) | **4** (1; *3*) | |
| | variety 18 | **1** (2; *33*) | | | | |
| | variety 19 | **1** (2; *6*) | **2** (1; *3*) | **3** (1; *3*) | **4** (1; *3*) | |
| 2020 | Italiano Classico | | **2** (2; *6*) | **3** (1; *3*) | **4** (2; *6*) | |
| | variety 1 | | **2** (1; *3*) | **3** (1; *3*) | **4** (1; *3*) | |
| | variety 3 | | **2** (1; *3*) | **3** (1; *3*) | **4** (1; *3*) | |
| | variety 5 | | **2** (1; *3*) | **3** (1; *3*) | **4** (1; *3*) | |
| | variety 6 | | | | **4** (1; *3*) | |
| | variety 9 | | | | **4** (1; *3*) | |
| | variety 10 | | | **3** (1; *3*) | | |
| | variety 12 | | **2** (1; *3*) | **3** (1; *3*) | **4** (1; *3*) | |
| | variety 14 | | **2** (1; *3*) | | **4** (1; *3*) | |
| 2021 | Italiano Classico | **1** (1; *3*) | **2** (1; *3*) | **3** (1; *3*) | **4** (1; *3*) | |
| | variety 2 | **1** (1; *3*) | **2** (1; *3*) | **3** (1; *3*) | **4** (1; *3*) | |
| | variety 4 | **1** (1; *3*) | **2** (1; *3*) | **3** (1; *3*) | **4** (1; *3*) | |
| | variety 8 | **1** (1; *3*) | **2** (1; *3*) | **3** (1; *3*) | **4** (1; *3*) | |
| | variety 9 | **1** (1; *3*) | **2** (1; *3*) | **3** (1; *3*) | **4** (1; *3*) | **5** (1; *3*) |
| | variety 11 | **1** (1; *3*) | **2** (1; *3*) | **3** (1; *3*) | **4** (1; *3*) | |
| | variety 12 | **1** (1; *3*) | **2** (1; *3*) | **3** (1; *3*) | **4** (1; *3*) | |
| | variety 14 | **1** (1; *3*) | **2** (1; *3*) | **3** (1; *3*) | **4** (1; *3*) | **5** (1; *3*) |
| | variety 15 | **1** (1; *3*) | **2** (1; *3*) | **3** (1; *3*) | **4** (1; *3*) | |
| | variety 16 | **1** (1; *3*) | **2** (1; *3*) | **3** (1; *3*) | **4** (1; *3*) | |

### 2.2. Sample Preparation and VOC Sampling

Samples of the basil plants were collected in the early morning, typically from 4 to 8 a.m., and rapidly sent to the lab for characterization. All samples were analyzed within 6–8 h from the cut to minimize deterioration. For the analysis, about 30 g of the whole basil plant (leaves and stems), exactly weighted with a precision of 0.1 g, were hashed in a blender (Oster, Sunbeam Products Inc., Boca Raton, FL, USA) for 30 s in 300 mL of extraction solution at room temperature. The extraction solution was 100 g $L^{-1}$ of NaCl and 6 mg $kg^{-1}$ of ethyl iso-butyrate in water. NaCl was added to increase the volatile molecules release in the extraction headspace and ethyl iso-butyrate was added as an internal standard for the fast GC analysis. After the 30 s blending step, the suspension was left for 30 s, then 20 µL was collected and transferred in 20 mL amber vials that were immediately sealed and sent to analysis. Each extract was sampled at least three times in different vials. All reagents, standard and solvents were of analytical grade (Sigma Aldrich, St. Louis, MO, USA).

### 2.3. Heracles E-Nose Fast-GC Analysis

The analysis of the volatile molecules in the sample headspace was carried out using a Heracles II (Alpha MOS, Tuluse, France) ultra-fast chromatography electronic nose [25]. The e-nose consists of a double-columns ultra-fast-chromatography system, with FID detectors, interfaced with a PAL-RSI automatic headspace autosampler. Sample headspace air was collected and injected in the e-nose. The injected air was trapped on a Tenax TA polymer trap positioned before the columns. The two columns are mounted in parallel in the oven and have different polarities, MXT-5 (non-polar) and MXT-1701 (slightly polar); both have a length of 10 m, internal diameter of 0.18 mm and a phase thickness of 0.40 µm. A temperature ramp was employed, starting from 50 °C for 2 s, then increasing to 80 °C at 1 °C/s and finally reaching 250 °C at 3 °C/s. The total fast GC analysis time was 110 s. The carrier gas was hydrogen.

Each replicate of the extracted samples was loaded in the instrument auto sampler and incubated for 20 min at 40 °C before injection with 500 rpm agitation (5 s on, 2 s off). Then, 1 mL of air headspace was injected with a syringe at the temperature of 50 °C. Trap loading conditions were 18 s at 40 °C, then flashed to 250 °C for the release in the two columns at a split ratio 1:1.

The AlphaSoft v 16.0 software was used for a preliminary process of the data that were subsequently exported for further elaborations.

Volatile compounds were putatively identified on the basis of Kovats' relative retention indices (KI) and can be related to specific aromas that are collected in the AroChemBase v 7.0 database (Alpha MOS, Tuluse, France) built-in software. In this way, eighteen compounds were tentatively identified, as reported in a previous work [21].

### 2.4. Data Analysis

#### 2.4.1. Data Preprocessing

Since the proper preprocessing of the different instrumental signals is very important to achieve trustworthy results, a preprocessing strategy was implemented to align the chromatograms.

The raw chromatograms, resulting from each of the two columns, were separately preprocessed as follows:

- First, they were normalized for the respective internal standard;
- Then, they were aligned by using the icoshift algorithm [26] applied by intervals, taking as reference the average signal. The intervals were manually defined, holding a single peak or small groups of peaks, as reported in Figure 1a. Alignment was necessary to compensate for the peaks shift, along retention time, among different chromatographic runs, which could introduce variability among samples not due to actual differences;

- The aligned chromatograms were baseline corrected by using the automatic weighted least squares algorithm (2nd order polynomial) [27];
- Considering that, in the analyzed chromatograms, the peaks' intensity and variance reflect the presence of major and minor constituents, it was important to use a procedure able to make the different chromatographic regions comparable in influence on the developed statistical models. In particular, block scaling to equal block variance (defining the blocks to be the same as the intervals used for the alignment with icoshift) was used, including column mean centering.

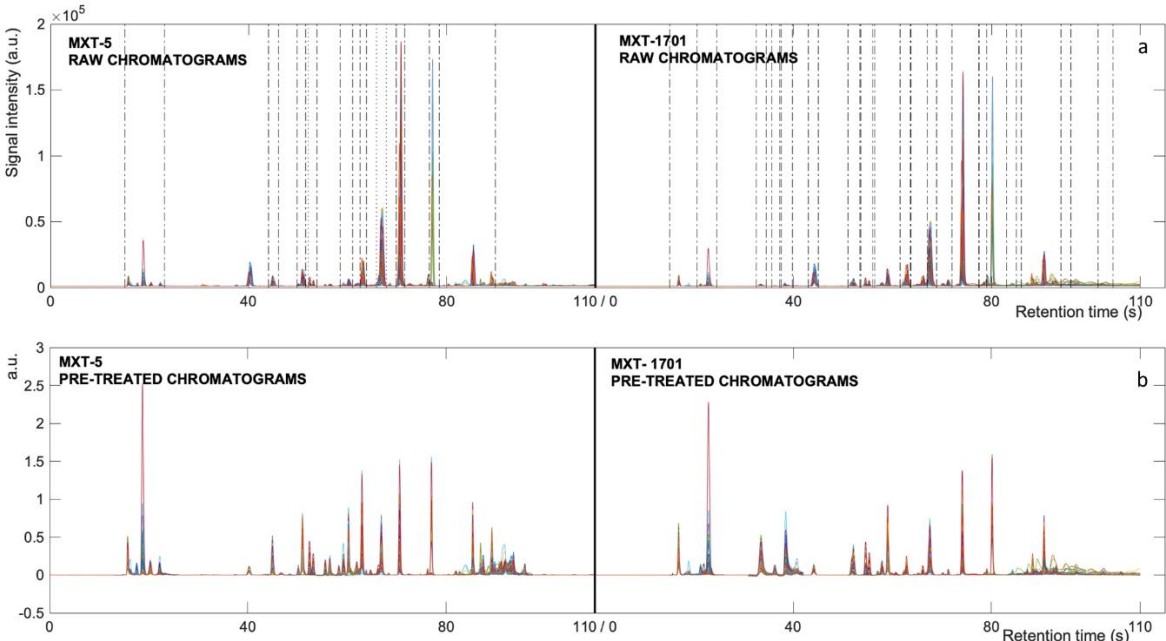

**Figure 1.** Collected chromatograms of basil samples (**a**) before and (**b**) after the different data pretreatments. (**a**) Dotted lines mark the limits of the different intervals used for the alignment and the scaling of the signals.

The preprocessed chromatograms are shown in Figure 1b.

A low-level data fusion approach was applied in order to simultaneously capture information coming from the analysis of samples through the two columns as well as to combine two potentially different sources of information. Indeed, from a chemical point of view, the slightly different polarity between the columns could highlight the presence of different analytes or obtain a better resolution, avoiding the loss of information due to possible co-elution issues. To this aim, the two singularly preprocessed chromatographic data sets were then concatenated in a single matrix of 352 (samples including replicates) × 20,002 (retention time points) dimensions. The MXT-5 and MXT-1701 chromatographic signals have a retention time ranging from 0 to 110 s sampled at 100 Hz, giving each 10,001 data points.

Prior to PCA, the concatenated data sets were block-scaled by considering as distinct data block each GC column (each data block comprises 10,001 variables, which are the respective sampled retention times), in order to let them equally contribute in PCA modelling.

### 2.4.2. ASCA

After data pretreatment (as detailed above in Section 2.4.1), the low-level fused chromatographic data (352 × 20,002 matrix dimensions) were subject to multivariate data analysis. As described in Table 1, samples varied according to three factors: harvesting year, variety and cut.

Principal Component Analysis (PCA) was applied on the entire data matrix to obtain a global overview of the trend, similarity and differences among the investigated samples according to the entire aroma profiles.

Furthermore, in order to assess the significance of the three factors (year, variety and cut) and their interactions, the ANOVA-Simultaneous Component Analysis (ASCA) method was used [24]. As a first step, ASCA performs an ANOVA, partitioning the data matrix X into the contribution of each factor or interaction, as shown in Equation (1):

$$X_c = X - 1m^T = X_1 + X_2 + X_3 + X_{1\times2} + X_{2\times3} + X_{1\times3} + X_{1\times2\times3} + X_{res} \tag{1}$$

where $X_c$ is the centered data matrix, $m^T$ is the mean profile of the samples, X (1, 2 and 3) is the main effect matrices, X ($1 \times 2$, $2 \times 3$, $1 \times 3$ and $1 \times 2 \times 3$) is the interaction effect matrices and $X_{res}$ is the residuals matrix. Then, a Simultaneous Component Analysis (SCA) is performed, obtaining a scores matrix T and a loadings matrix P for each effect or interaction matrix, as described by Equation (2):

$$X_i = T_i P_i^T \tag{2}$$

ASCA needs balanced designs to provide reliable results. In order to avoid the construction of a design where the number of combinations per factor level is not equal, 18 conditions were selected from the original dataset for a total of 54 experiments, as shown in Table 2. Thus, in this model, three levels for factor "year" (2019, 2020 and 2021), three for factor "variety" (Italiano Classico, VAR 9 and VAR 14) and two levels for factor "cut" (2 and 4) were considered.

**Table 2.** Structure of the experimental design for ASCA for the years 2019, 2020 and 2021.

| Year | Variety | Cut |
| --- | --- | --- |
| 2019 | Italiano Classico | 2 |
| 2019 | Italiano Classico | 4 |
| 2019 | VAR 9 | 2 |
| 2019 | VAR 9 | 4 |
| 2019 | VAR 14 | 2 |
| 2019 | VAR 14 | 4 |
| 2020 | Italiano Classico | 2 |
| 2020 | Italiano Classico | 4 |
| 2020 | VAR 9 | 2 |
| 2020 | VAR 9 | 4 |
| 2020 | VAR 14 | 2 |
| 2020 | VAR 14 | 4 |
| 2021 | Italiano Classico | 2 |
| 2021 | Italiano Classico | 4 |
| 2021 | VAR 9 | 2 |
| 2021 | VAR 9 | 4 |
| 2021 | VAR 14 | 2 |
| 2021 | VAR 14 | 4 |

Moreover, in order to further investigate the influence of varieties and cuts on basil aromatic profiles, another ASCA model was computed considering the year 2021 (where a higher number of varieties was cultivated), giving the sub-set of experiments described in Table 3. In this case, 9 basil varieties and 3 different cuts were inspected, for a total of 27 conditions and 81 experiments. It was not possible to investigate all levels for each

experimental factor, due to the limited varieties available that could be cultivated by a single producer.

**Table 3.** Structure of the experimental design for ASCA for the year 2021.

| Variety | Cut |
|---|---|
| Italiano Classico | 1 |
| Italiano Classico | 2 |
| Italiano Classico | 4 |
| VAR 2 | 1 |
| VAR 2 | 2 |
| VAR 2 | 4 |
| VAR 4 | 1 |
| VAR 4 | 2 |
| VAR 4 | 4 |
| VAR 8 | 1 |
| VAR 8 | 2 |
| VAR 8 | 4 |
| VAR 9 | 1 |
| VAR 9 | 2 |
| VAR 9 | 4 |
| VAR 12 | 1 |
| VAR 12 | 2 |
| VAR 12 | 4 |
| VAR 14 | 1 |
| VAR 14 | 2 |
| VAR 14 | 4 |
| VAR 15 | 1 |
| VAR 15 | 2 |
| VAR 15 | 4 |
| VAR 16 | 1 |
| VAR 16 | 2 |
| VAR 16 | 4 |

The significance of the effect of each design factor or interaction was evaluated through permutation tests (1000 randomizations), which compared the experimental sum of squares of each effect matrix with its related distribution under the null hypothesis [28].

2.4.3. Software

The raw chromatograms were imported and processed under a MATLAB 2020a (The MathWorks, Inc., Natick, MA, USA) environment. Chromatogram alignment was performed by using the icoshift 3.0, freely available on www.models.kvl.dk (last access on 7 March 2021). PCA and preprocessing were performed by PLS-Toolbox v. 8.9 (Eigenvector Inc., Manson, WA, USA). ASCA was carried out by using routines developed and kindly made available by Dr. F. Marini, University of Roma La Sapienza (Italy).

## 3. Results and Discussions

### 3.1. PCA Exploratory Analysis

In this first exploratory analysis, the aim was to obtain a general overview of the variation of the aroma volatile fraction of basil samples. Punctual considerations of the influence of harvested year, variety and cut could not be conducted, since it was not possible to plain a systematic sampling beforehand, due to company and producer constrains. Three principal components were considered according to their explained variances (58%). In Figure 2, the PC1 vs. PC2 score plot is reported, representing the different basil samples with different symbols and color as function of harvesting year and basil variety (Figure 2a) or cut and basil variety (Figure 2b).

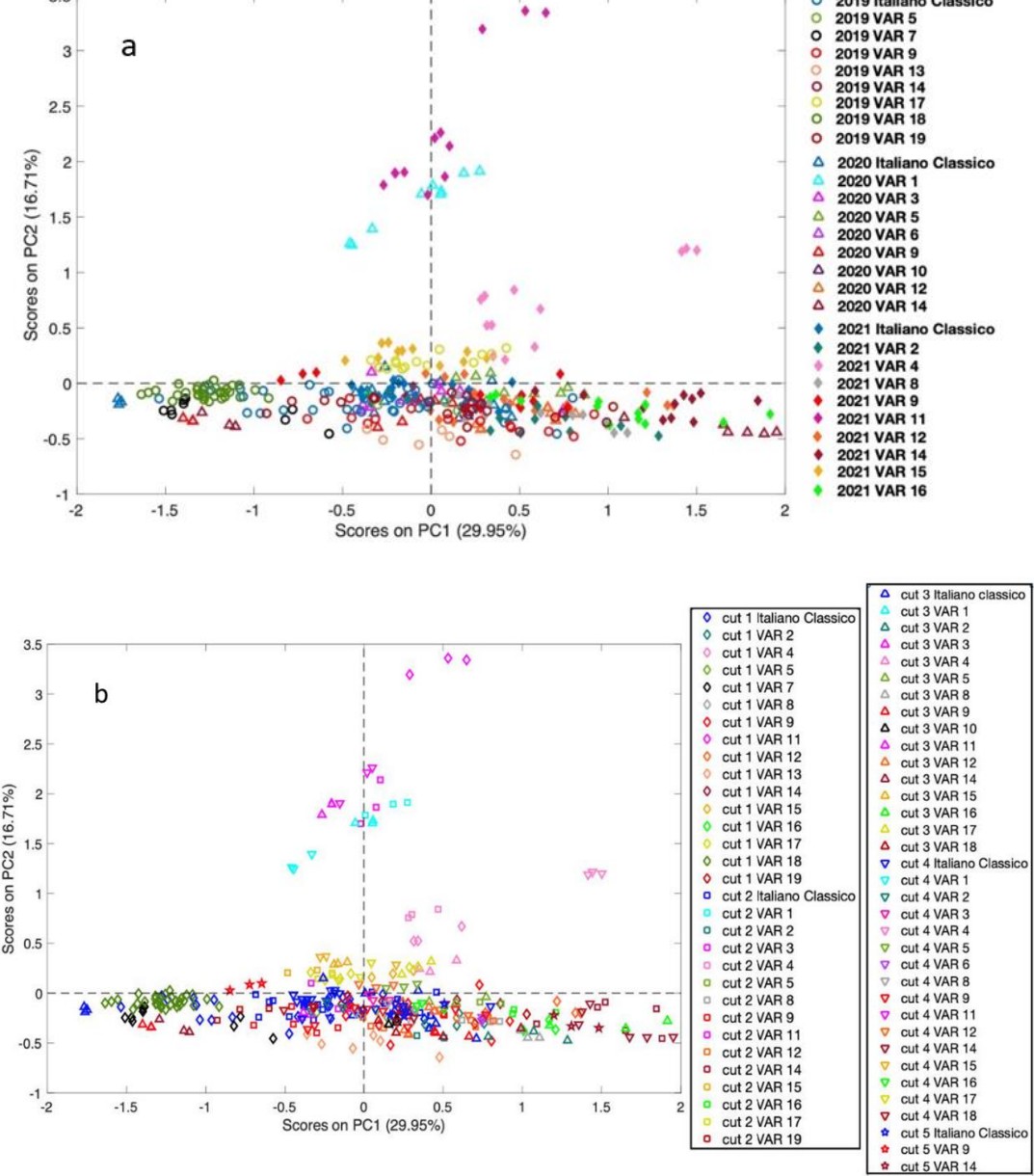

**Figure 2.** PC1 vs. PC2 score plots of basil samples. (**a**) Different symbols are used for each harvesting year (2019: circles; 2020: squares; 2021: triangles) and different colors for each basil variety. (**b**) Different symbols are used for each cut (first: diamonds; second: squares; third and fourth: upwards and downwards triangles, respectively; fifth: stars) and different colors for each basil variety.

From the score plot of the first two components, it is difficult to highlight a clear separation of samples according to all the different basil varieties, due to the slight differences in the flavor pattern among commercial varieties that belong to the same species (*O. basilicum*). However, interesting information can be pointed out. In particular, the VAR 1 (harvested only in 2019) and VAR 11 (harvested only in 2021) samples have the highest PC2 score values and leads to their separation from the other samples (Figure 2a). These varieties also present a trend, from higher to lower score values, according to their different cut (Figure 2b). Another peculiar variety seems to be VAR 4 (harvested only in 2021), with positive scores for both PC1 and PC2. This variety shows differences in aroma according to different basil cuts as well.

As far as the other samples are concerned, they are distributed along the first principal component, which seems to be the most responsible for the differences in the separation between the VAR 14 samples (higher positive PC1 score values) and first cut of VAR 7, VAR 18 and Italiano Classico (negative PC1 score values).

Furthermore, the in-depth analysis of the figure shows that two samples belonging to the third cut of VAR 16 (higher PC1 score values) seem to have quite a similar aroma profile to VAR 14.

No further observations to assess any pattern can be performed considering the different basil cuts, years and varieties, since it is not certain what the real cause is as some varieties were measured only in one year.

The score plot of the third component (Figure S1 reported in Supplementary Material) highlights the differences among the first basil cut of the VAR 8 and VAR 17 samples (higher positive score values) with respect to all the others.

From the PC1 loading plot (Figure 3a), for both MXT5 and MXT17 columns, it is possible to point out that, with almost all the loadings values being positive (from 40 to 110 s), the separation between the VAR 14 samples and the other basil varieties is mainly due to a global higher concentration of aroma compounds in these samples, and roughly speaking, most of the samples harvested in 2021 (positive PC1 score values) seem to present a similar trend.

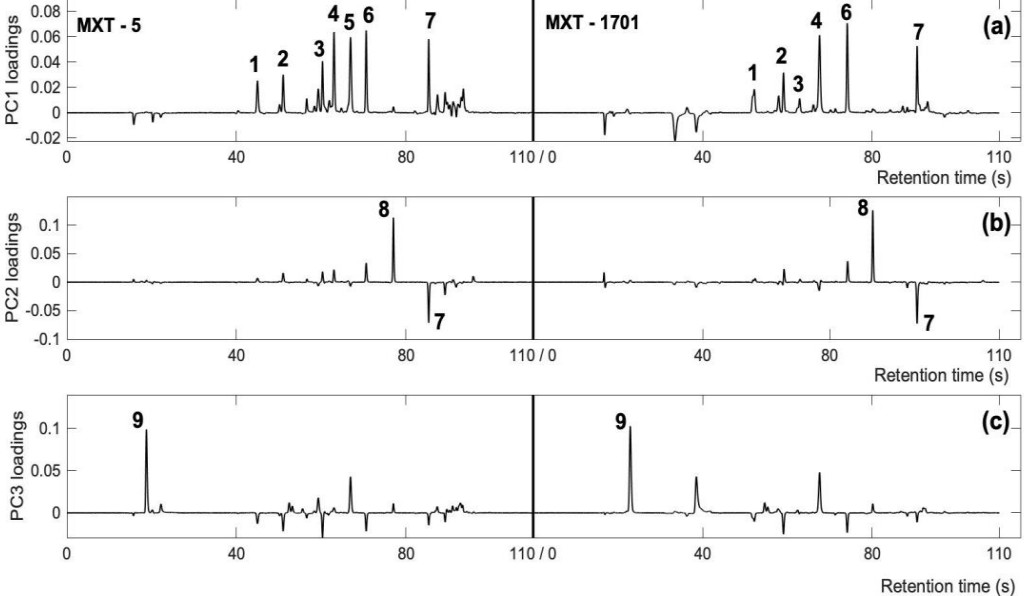

**Figure 3.** (**a**) PC1, (**b**) PC2 and (**c**) PC3 loading plots. Numbered peaks correspond to the volatile compounds putatively identified on the basis of Kovats's relative retention indices: (1) hexanal, (2) 2-hexanal, (3) 5-methylfurfural, (4) myrcene, (5) eucalyptol, (6) linalool, (7) β-caryophyllene, and (8) eugenol (9) not identified.

Notwithstanding the aim of the present study, which is to make a fast model to discriminate basil samples with an untargeted approach, some considerations on the presence of some chemical compounds can be presented on the basis of our previous study. Regarding the second principal component (Figure 3b), which is mainly responsible for the separation of VAR 1 and VAR 11 from the others, the same chromatographic regions (Rt, retention time: 76.8 s and 85.3 s for MXT-5 and 79.9 s and 90.4 s for MXT-17), for both the MXT-5 and MXT-17 columns, with the same trend (loadings value and sign), are relevant. Thus, both the estragole (Rt: 76.8 s and 79.9 s in MXT-5 and MXT-1701, respectively) and eugenol compounds (Rt: 85.3 s and 90.4 s in MXT-5 and MXT-1701, respectively), with high positive and negative loading values, respectively, are important to characterize VAR 1 and VAR 11. However, the samples belonging to these two varieties, presented a particular aroma, probably due to the presence of anethole, which co-elutes with estragole in both column separations.

As regards the third principal component (Figure 3c), unassigned compounds (in the first 40 s of both columns), which have positive loadings, seem more abundant in the VAR 8 and VAR 17 samples (located at positive scores values). Hence, further investigation will be conducted for the identification of these volatile compounds.

Notwithstanding the overall interpretation of PCA results, which offered some insights, more specific information is difficult to gain, since the contributions to variance of all the investigated factors (i.e., year, variety and cut) overlap. Therefore, after this preliminary investigation, the ASCA methodology was used in order to systematically assess the influence of each factor and their interaction on the basil aroma profile.

### 3.2. ASCA Results

The first ASCA model was computed according to the experimental design scheme shown in Table 2 (Section 2.4.2). The original data matrix variation was split in eight submatrices: three corresponding to the main effect of each experimental factor, three accounting for the effect of each second-order interaction, one describing the effect of the third-order interaction and one holding the residuals. The significance of all these effects was assessed by performing a permutation test, whose results are shown in Table 4. The *p*-value of all the inspected factors and interactions was lower than 0.001. However, the factors "variety" and "year" explained most of the data variance (39.9% and 24.8%, respectively), suggesting their higher influence on the aromatic profile of basil compared to the factor "cut". This can also be observed by the fact that explained variance values of interactions including "cut" are systematically lower than values related to interactions in which "cut" is not involved. Additionally, the third-order interaction effect explains less than 3% variance.

**Table 4.** Explained variance and *p*-values for main factors and their second and third order interactions.

| Factor | Explained Variance (%) | *p* |
|---|---|---|
| Variety | 39.9 | <0.001 |
| Year | 24.8 | <0.001 |
| Year x Variety | 8.5 | <0.001 |
| Year x Cut | 7.2 | <0.001 |
| Cut | 2.9 | <0.001 |
| Variety x Cut | 2.5 | <0.001 |
| Year x Variety x cut | 2.8 | <0.001 |

Afterwards, the ASCA algorithm performed a SCA on each effect matrix individually, with the aim of interpreting the observed variation.

Figure 4a shows the score plot for the factor "year". The first component (SC1), which explains 67.7% of the total variance, describes the difference between the samples

harvested in 2019 and the samples harvested in 2020 and 2021. The loadings plot of the first component, shown in Figure 4b, explains this difference. In fact, the 2020 and 2021 samples appear to have a richer aroma profile, as the concentration of the compounds between 40 and 110 s, associated with statistically significant loadings, are higher compared to 2019 samples. On the other hand, 2019 samples present higher concentrations of unassigned peaks before 40 s mainly highlighted by the MXT-1701 column, confirming the need of further investigation for their identification.

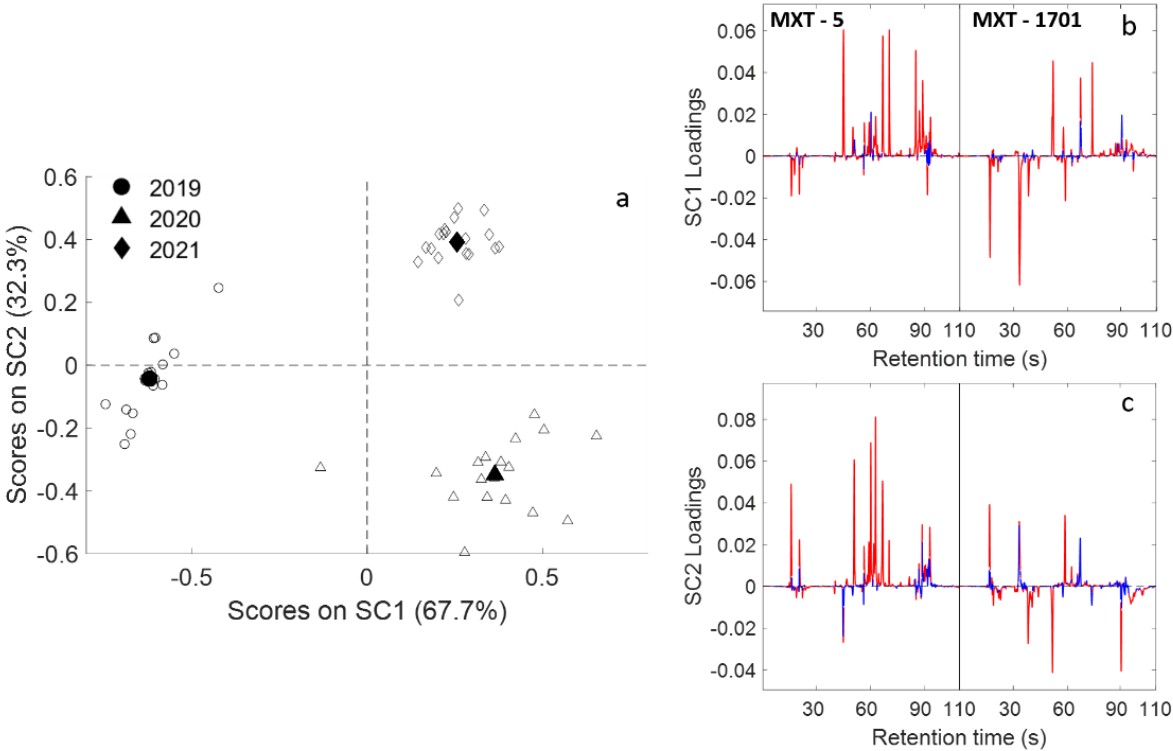

**Figure 4.** SCA for the effect of the factor "year". (**a**) SC1 vs. SC2 score plot. Empty symbols represent the projected residuals; (**b**) SC1 and (**c**) SC2 loadings plot. In the loading plots, red lines indicate statistically significant regions, whereas blue lines indicate regions associated with loadings statistically indistinguishable from zero.

The second component (SC2) and the related loadings plot (Figure 4c) show how the 2021 samples (positive scores values) present lower peaks in MXT-1701 that can be ascribed to 2-hexanal and β-caryophyllene (negative loadings values), but higher peaks assigned to all other compounds.

Figure 5a shows the score plot for the factor "variety". It can be observed that most of the explained variance (96.3%) describes how VAR 14 is different compared to Italiano Classico and VAR 9. Indeed, as shown by the loadings plot in Figure 5b, VAR 14 presents higher concentrations of all the chromatographic peaks, suggesting a richer aroma profile with respect to the other two varieties. SC2, even though the related explained variance is very low (3.7%), mainly shows how VAR 9 has more β-caryophyllene than Italiano Classico (Figure 5c), as their peaks are basically the only ones that had statistically significant results.

The results of the SCA for the effect of the interaction "year x variety" are reported in Figure 6. In the score plot (Figure 6a), it can be observed that SC1 describes the difference among VAR 14 samples throughout the years. In detail, the VAR 14 samples collected in 2020 presented a higher concentration of all aroma compounds compared to the ones collected in 2019 and 2021, as assumed by the loadings plot shown in Figure 6b. As regards Italiano Classico, the best year in terms of intensity of aroma profile is 2019, whereas for VAR 9, the years 2019 and 2021 were better than 2020.

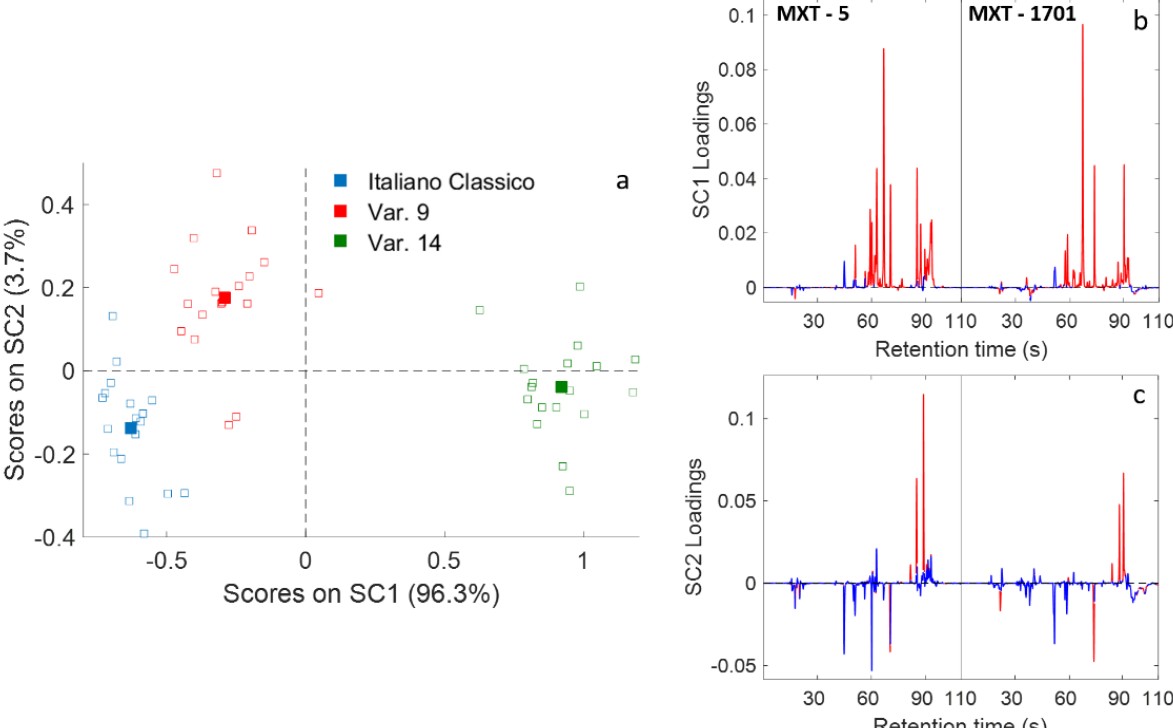

**Figure 5.** SCA for the effect of the factor "variety". (**a**) SC1 vs. SC2 score plot. Empty symbols represent the projected residuals; (**b**) SC1 and (**c**) SC2 loadings plot. In the loading plots, red lines indicate statistically significant regions, whereas blue lines indicate regions associated with loadings statistically indistinguishable from zero.

It can also be observed how VAR 14 appears to change more over time, having a higher variation through the years than the other two varieties.

Moreover, Italiano Classico is the basil variety that presents the lowest variability among its replicates. In fact, red and green samples in the score plot (VAR 9 and VAR 14, respectively) are more spread and farther apart, especially along SC2. This limits further comments about the difference between the years 2020 and 2021 with respect to the Italiano Classico samples (blue triangles and diamonds in Figure 6a, respectively), which is due to the statistically significant peaks between 50 and 70 s, linked to the majority of the aromatic compounds.

Regarding the factor "cut", the SCA showed how samples collected during cut 2 detain a richer aroma profile than samples acquired during cut 4. However, according to the authors, since this factor explained less than 3% of the total variance, these results are not relevant compared to the ones described above. Both for this reason and for the sake of brevity, plots related to the factor "cut" were not shown.

The second ASCA model was computed taking into account only samples collected in 2021. In this case, it was possible to build a balanced design, including nine varieties and three cuts, according to the scheme shown in Table 3 (Section 2.4.2). The data matrix was partitioned in four submatrices: two corresponding to the main effect of each experimental factor, one describing the effect of the second-order interactions and the residuals matrix. The results of the permutation test for the significance of the effects are shown in Table 5. As for the first ASCA model, also in this case, all the factors and their interactions were significant ($p < 0.001$). Furthermore, the explained variance for the factor "cut" (6.9%) was significantly lower than the variance explained by the factor "variety" (63.5%), suggesting, once again, the small impact of plant age on the basil aroma profile.

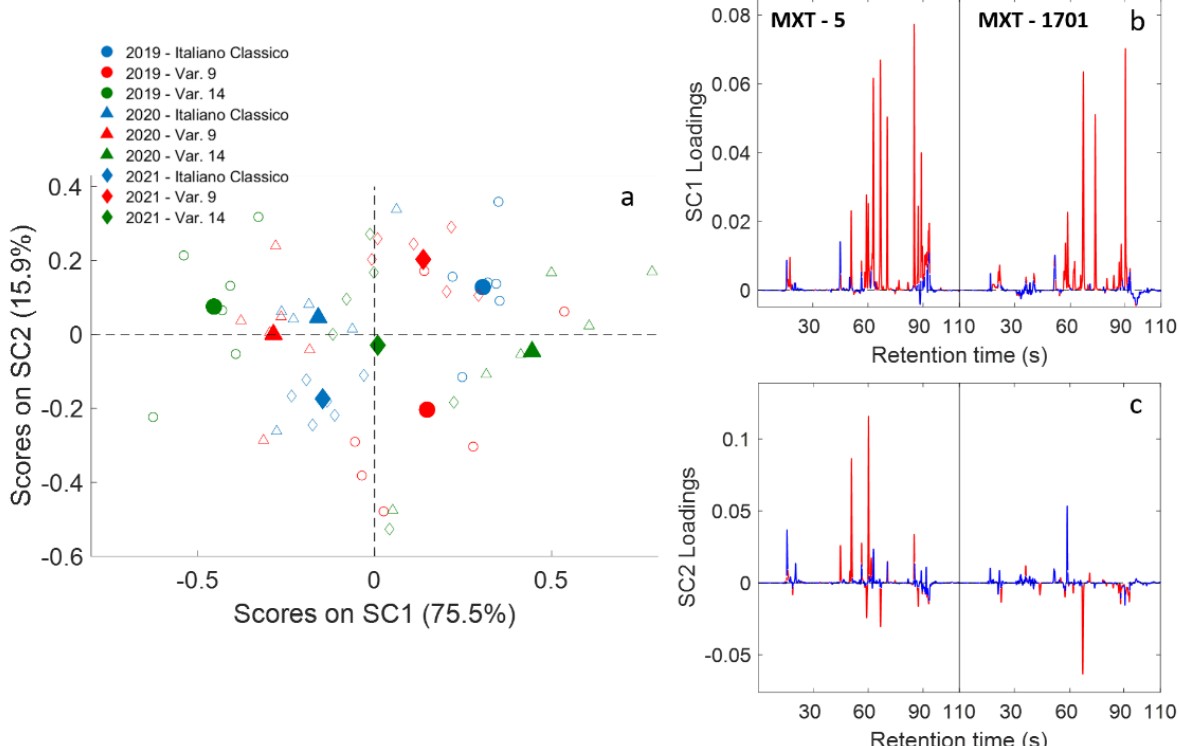

**Figure 6.** SCA for the effect of the interaction "year x variety". (**a**) SC1 vs. SC2 Score plot. Empty symbols represent the projected residuals; (**b**) SC1 and (**c**) SC2 loadings plot. In (**a**), the different colors refer to the different varieties (blue—Italiano Classico; red—VAR 9; green—VAR 14), whereas different symbols refer to different harvesting years (circles—2019; triangles—2020; diamonds—2021). In loading plots, red lines indicate statistically significant regions, whereas blue lines indicate regions associated with loadings statistically indistinguishable from zero.

**Table 5.** Explained variance and *p*-values for main factors and their second order interactions related to the ASCA model.

| Factor | Explained Variance (%) | *p* |
|:---:|:---:|:---:|
| Variety | 63.5 | <0.001 |
| Variety x Cut | 20.3 | <0.001 |
| Cut | 6.9 | <0.001 |

The results related to the SCA on the "variety" effect matrix are shown in Figure 7.

From the score plot (Figure 7a), it is clear how the first principal component shows the difference between VAR 4 and all the other varieties. In the loadings plot (Figure 7b), it is shown that the peak that is mainly responsible for this difference can be ascribed to myrcene, of which VAR 4 is particularly rich. Observing SC2 scores and loadings (Figure 7c), it can be concluded that VAR 14 and VAR 16 present the richest aroma profiles, whereas Italiano Classico and VAR 15 have the poorest profiles.

Figure 8a shows the frequency histogram of the SC1 scores values for the different levels of the factor "cut". Eucalyptol and β-caryophyllene are less present in cut 4 samples, and in general, they are the compounds responsible for describing the difference between cut 4 samples and cut 1 and 2 samples, as shown in Figure 8b.

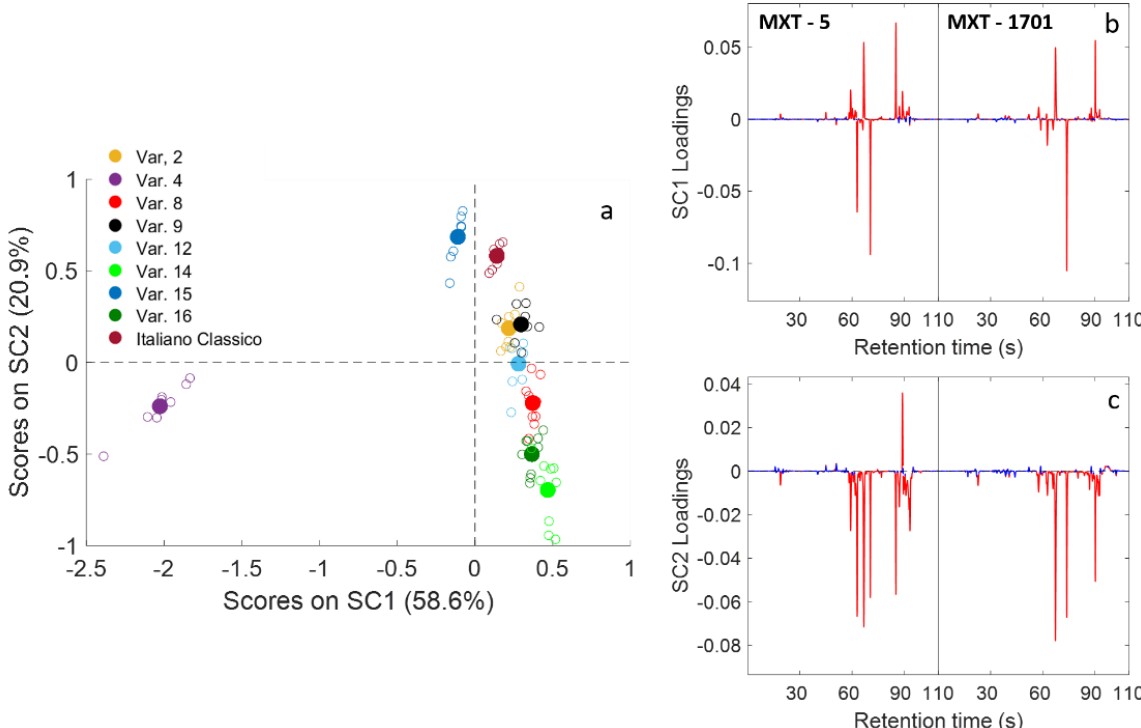

**Figure 7.** Results of ASCA performed on the 2021 samples. The SCA for the effect of the factor "variety". (**a**) SC1 vs. SC2 score plot. Empty symbols represent the projected residuals; (**b**) SC1 and (**c**) SC2 loadings plot. In loading plots, red lines indicate statistically significant regions, whereas blue lines indicate regions associated with loadings statistically indistinguishable from zero.

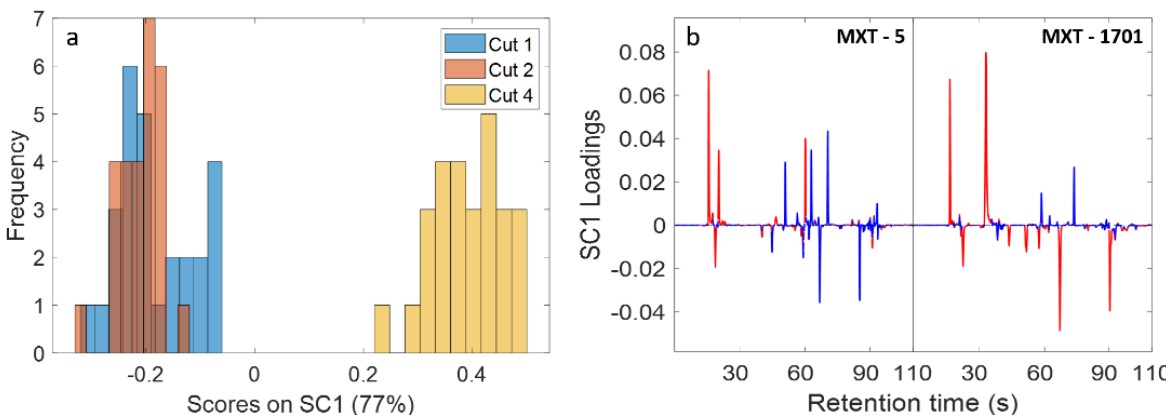

**Figure 8.** Results of ASCA performed on 2021 samples. The SCA for the effect of the factor "cut". (**a**) histograms of ASCA score frequency (with projected residuals) on SC1 for the different levels of factor "cut"; (**b**) SC1 Loadings plot. In loading plots, red lines indicate statistically significant regions, whereas blue lines indicate regions associated with loadings statistically indistinguishable from zero.

The ASCA results show how the entire aromatic profile has a significant influence in the discrimination of samples according to the investigated factors (i.e., years, variety and cut), highlighting the presence of new potential biomarkers (for instance the species with retention time in the first 30 s of the chromatogram or the ones falling in the area between the retention of 2-hexanal and 5-methylfurfural), which have not been quantified in this study, but that could be relevant in further investigations. For the sake of clarity, an example signal fingerprint with all the chemical analytes, putatively identified for both the chromatographic separations, is reported in Figure 9.

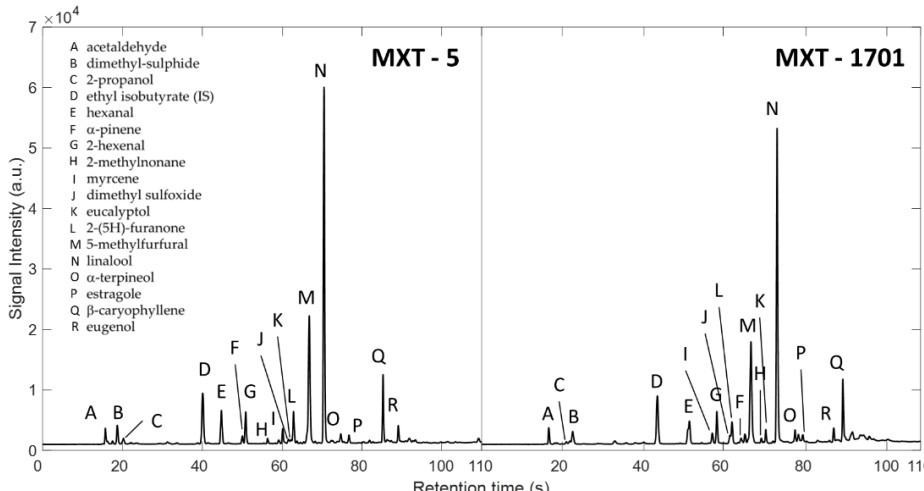

**Figure 9.** Chromatograms of the Italiano Classico variety obtained by elution on columns MXT-5 and MXT-1701 of Heracles II.

## 4. Conclusions

In this study, the development of a fast analytical screening strategy based on an ultra-fast chromatography e-nose and multivariate analysis was proposed as a useful tool for quality control of food. The proposed approach, relying on the simultaneous analysis of the chromatographic profiles coming from two GC-columns of different polarity, permits to explore fully the volatile profile of foodstuff and may represent a fast and simpler alternative to other chromatographic techniques. The chemical identification and quantification of the single chemical species, responsible for differentiation of the studied food products, can be undertaken on a few samples for a second time. In fact, once the main chromatographic peaks, mostly responsible for the differentiation between samples, have been underlined, their respective chemical species can be identified with a considerable reduction in costs and analysis time.

In particular, this approach was applied on the analysis of the basil samples involved in the production of Italian pesto sauce, where the entire e-nose signals, coming from two columns with different polarity, were fused and used as a fingerprint of the aroma profile. The obtained results highlighted the possibility of differentiating basil samples on the basis of the three investigated factors, years, cut and variety, taking also into account the interactions among them. The low-level data fusion approach allowed the computing of a single ASCA model, which effectively pointed out the different significant peaks between the two columns taken into account, thus underlining that enhanced information may be gained.

The knowledge of the influence of the investigated factors on the quality of basil is very important, since it may allow a company to achieve useful information both to plan future campaign strategies for the acquisition of the raw materials and to improve the quality of the final pesto sauce.

**Supplementary Materials:** The following supporting information can be downloaded at: https://www.mdpi.com/article/10.3390/chemosensors10030105/s1, Figure S1: PC3 scores vs. n° of sample.

**Author Contributions:** Conceptualization, M.C., C.D. and A.D.; methodology, M.C., A.D., D.B., C.D. and L.S.; software, C.D. and L.S.; validation, M.C., A.D. and C.D.; investigation, A.D., D.B., C.D. and L.S.; resources, A.D.; data curation, A.D., C.D. and L.S.; writing—original draft preparation, A.D., C.D., and L.S.; writing—review and editing, M.C., A.D., C.D. and L.S.; supervision, M.C. and C.D.; project administration, A.D.; funding acquisition, A.D. All authors have read and agreed to the published version of the manuscript.

**Funding:** This research received no external funding.

**Institutional Review Board Statement:** Not applicable.

**Informed Consent Statement:** Not applicable.

**Data Availability Statement:** The data are available on request from the authors.

**Acknowledgments:** The authors acknowledge Flavio Bertinaria for supplying basil samples and Federica Quaini for the sensorial descriptions.

**Conflicts of Interest:** Authors A.D. and D.B. are employed by Barilla G. e R. Fratelli SpA. The remaining authors declare that the research was conducted in the absence of any commercial or financial relationships that could be construed as a potential conflicts of interest.

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
