# Peer review of "Fast GC E-Nose and Chemometrics for the Rapid Assessment of Basil Aroma"

_chemosensors, doi:10.3390/chemosensors10030105_

Round 1

Reviewer 1 Report

In the submitted manuscript the authors present an analysis of experimental results of basil aroma measurements using gas chromatography-based electronic nose.
Such a method is promising as it allows not only to find patterns indicating differentiation between measured samples but also to identify constituting chemical components.

The authors present an analysis in which samples odors dependence on three factors is explained: year, variety, and cut.
However, in my opinion, there are some flaws in the description of the analytical procedure, and some conclusions are perhaps not fully justified.
As we can notice in Figure 1, where it is presented what samples were used for measurements, not all combinations of these factors are available.
I think the most important is that some varieties were measured only in one year. 
As we can see in Figure 2, probably all experimental data points are used, and then when one tries to assess any pattern in these data it is not sure what is the real cause.
For example, when we examine figure 2c and see some outliers it is not sure if they are caused by the year factor or maybe they are points representing some variety that was measured only in this particular year. As we can notice in Figure 2a, outliers are belonging to a particular variety, and also a pattern can be noticed, that data points for a given variety form close clusters. But also it should be noticed, that not only outliers in Figure 2c may be caused by variety, which is probably the dominant factor, but also the whole pattern in this Figure may be caused by the fact that varieties measured in each year are different and only a few were measured in all years. 
In my opinion analysis and discussion on the year factor are not sufficiently supported by data.
The same objection could be raised for data presented in Figure 2b. It is not sure if outliers points presented here are not data belonging to one or two distinct varieties and it is the real cause of this pattern.

More viable analyses, for which only 3 varieties were chosen, or analyses limited to data collected in the 2021 year, are more viable.

My advice to the authors is to use a visualization method that would allow them to present in one figure more factors, for example, using symbols for one factor and color for another. Maybe such visualization could help to reveal more interesting patterns or confirm/exclude some of the discussed findings. The authors used such different symbols and colors in Figure 6.

In Figure 6 there is no legend and the only description of the meaning of symbols and colors in the figure caption. It would be better to add legend as in other figures.

In line 297 the authors mention some more results not presented in the manuscript. Maybe the authors should consider including them as supplementary materials.

Reviewer 2 Report

Authors used fast GC e-nose for the evaluation of basil aroma. For that purpose, low level data fusion approach was used, combining the responses obtained when two different GC columns were used. My main concern is the novelty of this study in relation to the study published in Reference 21, bearing in mind that new compounds were not detected in this one and the same biomarkers monitored in the previous study were used in this one.

Secondly, the advantage of data fusion should be highlighted in the manuscript.

Other comments:

  • Abstract, line 15: “Anova-Simultaneous Component Analysis (ASCA)”.
  • Section 2.1. Indicate the total number of analysed samples as well as the number of replicates per sample.
  • Figure Caption 1: There are four figures, but clarify which one corresponds to Figure 1a or 1b.
  • A new figure showing the fingerprint signal could be added.

Finally, check English version of the manuscript. For instance, it contains several typos: e. g. line 56 “alchools”; line 126 “weighted”, etc….

Round 2

Reviewer 1 Report

I think there is a mistake in the preparation of Figure 2b.
In figure 2a the authors differentiate by various symbols year and by a various color variety of the samples.
In figure 2b they use various symbols and colors to differentiate cut. In my opinion, the most important factor is variety so maybe the authors should prepare this figure using different colors and different symbols to visualize both dimensions - cut and variety.

Reviewer 2 Report

Authors have addressed the issues indicated by former reviewers. However, English version of the manuscript should be revised and typos must be corrected. Some examples: line 101 "knoledge"; line 138 "CG"; line 474 "istance";...
